# Apparent Diffusion Coefficient Map-Based Texture Analysis for the Differentiation of Chromophobe Renal Cell Carcinoma from Renal Oncocytoma

**DOI:** 10.3390/diagnostics12040817

**Published:** 2022-03-26

**Authors:** Yusuke Uchida, Soichiro Yoshida, Yuki Arita, Hiroki Shimoda, Koichiro Kimura, Ichiro Yamada, Hajime Tanaka, Minato Yokoyama, Yoh Matsuoka, Masahiro Jinzaki, Yasuhisa Fujii

**Affiliations:** 1Department of Urology, Tokyo Medical and Dental University Graduate School, 1-5-45 Yushima, Bunkyo-Ku, Tokyo 113-8510, Japan; uchida1986@gmail.com (Y.U.); m22.nameko@gmail.com (H.S.); hjtauro@tmd.ac.jp (H.T.); mntykym.uro@tmd.ac.jp (M.Y.); yoh-m.uro@tmd.ac.jp (Y.M.); y-fujii.uro@tmd.ac.jp (Y.F.); 2Department of Radiology, Keio University School of Medicine, 35 Shinanomachi, Shinjuku-Ku, Tokyo 160-8582, Japan; yarita@rad.med.keio.ac.jp (Y.A.); jinzaki@rad.med.keio.ac.jp (M.J.); 3Department of Diagnostic Radiology, Tokyo Medical and Dental University Graduate School, 1-5-45 Yushima, Bunkyo-Ku, Tokyo 113-8510, Japan; kmrdrnm@tmd.ac.jp (K.K.); yamadachr@gmail.com (I.Y.)

**Keywords:** oncocytoma, renal cell carcinoma, diffusion magnetic resonance imaging, machine learning

## Abstract

Preoperative imaging differentiation between ChRCC and RO is difficult with conventional subjective evaluation, and the development of quantitative analysis is a clinical challenge. Forty-nine patients underwent partial or radical nephrectomy preceded by MRI and followed by pathological diagnosis with ChRCC or RO (ChRCC: *n* = 41, RO: *n* = 8). The whole-lesion volume of interest was set on apparent diffusion coefficient (ADC) maps of 1.5T-MRI. The importance of selected texture features (TFs) was evaluated, and diagnostic models were created using random forest (RF) analysis. The Mean Decrease Gini as calculated through RF analysis was the highest for mean_ADC_value. ChRCC had a significantly lower mean_ADC_value than RO (1.26 vs. 1.79 × 10^−3^ mm^2^/s, *p* < 0.0001). Feature selection by the Boruta method identified the first-quartile ADC value and GLZLM_HGZE as important features. ROC curve analysis showed that there was no significant difference in the classification performances between the mean_ADC_value-only model and the Boruta model (AUC: 0.954 vs. 0.969, *p* = 0.236). The mean ADC value had good predictive ability for the distinction between ChRCC and RO, comparable to that of the combination of TFs optimized for the evaluated cohort. The mean ADC value may be useful in distinguishing between ChRCC and RO.

## 1. Introduction

It is a clinically significant problem that a certain number of benign renal mass are surgically resected under the clinical diagnosis of renal cell carcinoma (RCC). Bauman et al. have reported that 14.1% of cases in which partial nephrectomy is performed under a diagnosis of RCC are pathologically benign [1]. Renal oncocytoma (RO) accounts for 5–7% of renal epithelial neoplasms and is the most common benign renal mass in Western countries [2,3]. Chromophobe RCC (ChRCC), in contrast, is a malignant tumor with metastatic potential, accounting for 6–8% of renal epithelial neoplasms [4]. Both ChRCC and RO are thought to arise from mesenchymal cells of the cortical collecting duct epithelium and have similar radiological and pathological features [5]. Therefore, the preoperative differential diagnosis of ChRCC and RO is one of the greatest diagnostic challenges related to renal tumors. Although percutaneous needle renal biopsy is an established diagnostic method for renal tumors, a previous study showed that about one-fourth of suspected RO cases by renal biopsy were diagnosed with RCC by surgical pathology, suggesting that it can be unreliable in differentiating ChRCC from RO [6]. Advances in radiographic analysis are desired, especially as a means of reducing the invasiveness of diagnosis.

Diffusion-weighted imaging (DWI) is a functional imaging technique that assesses the restriction of free water movement in tissues. The apparent diffusion coefficient (ADC) value, a quantitative parameter of the DWI signal, has been reported to be useful in assessing the biological and histological aggressiveness of various types of tumors, including renal tumors, and can be an imaging biomarker that reflects certain characteristics of the tumor [7,8,9]. Previous studies have suggested that the mean ADC value, one of the imaging features obtained from ADC maps, may be useful in distinguishing between ChRCC and RO [10,11,12]. To the best of our knowledge, however, no study has yet evaluated the usefulness of comprehensively analyzed image features of ADC maps.

In recent years, radiomics has attracted much attention for its potential to support clinical decision making and outcome prediction by enabling the analysis of large numbers of quantitative image features extracted from medical images [13,14]. Texture analysis (TA) provides a comprehensive assessment of the spatial heterogeneity of the signal in a tumor, and machine learning can make detailed predictions of tumor characteristics from the large number of data extracted. The aim of this study was to evaluate the usefulness of ADC-map-based TA in differentiating between ChRCC and RO.

## 2. Materials and Methods

### 2.1. Patient and Tumor Characteristics

This study, a retrospective single-institution analysis, was approved by our institutional review board (approval number: M2018-247-01), and written informed consent was obtained from each participant. Of the 55 patients who underwent partial or radical nephrectomy and were pathologically diagnosed with RO or ChRCC between 2011 and 2020, 49 patients (ChRCC: *n* = 41, RO: *n* = 8) who had been preoperatively evaluated by the following MRI protocol were included in this study. All of the tumors in the included patients were solitary.

### 2.2. MRI Settings

In the present study, we performed MRI using a 1.5T MR imager (Intera Achieva; Philips, Best, The Netherlands) with a 32-channel sensitivity-encoding body coil under free-breathing conditions. The maximal gradient strength was 33 mT/m and the slew rate was 160 T/m/s. Following routine T1- and T2-weighted imaging (T1W and T2W, respectively), DWI was performed. The imaging parameters of DWI with a single-shot echo planar imaging sequence were set as follows: repetition time: 1500 ms; echo time: 65 ms; matrix: 160 × 160; field of view: 360 mm × 240 mm; slice thickness: 7 mm; interslice gap: 0.7 mm; number of excitations: 3; and fat suppression: spectral presaturation inversion recovery. The diffusion gradient for the DWI procedure encoded three orthogonal directions, and the b values were 0, 500, and 1000 s/mm^2^. The total acquisition time was approximately 100 s. The ADC maps of the tumors were reconstructed at a workstation (Philip View Forum R4.1; Philips).

### 2.3. MRI Qualitative Analysis

The following conventional qualitative features were documented: size in the longest dimension, T1- and T2-weighted signal intensity, T2 homogeneity, tumor margin type (pseudo capsule), central scar (denoted by a central area of T2 hypointensity), cystic component, calcification, hemorrhage (denoted by high signal intensity on T1-weighted image), and fat (denoted by a drop in signal intensity on opposed-phase image).

### 2.4. Texture Feature Analysis

We performed the extraction and calculation of texture features using LIFEx software v5.10 (IMIV, CEA, Orsay, France) [15]. An overview of the workflow is depicted in Figure 1. For all slices of each renal tumor, a maximum region of interest (ROI) was set at the tumor, and the software semi-automatically delineated a three-dimensional region of interest (i.e., a volume of interest or VOI). After the spatial resampling step, radiographic features of 2.0 × 2.0 × 2.0 mm voxel size were calculated. The number of intensity levels was resampled using 64 discrete values (64 bins) between the absolute minimum and maximum values in the VOI. Any tumor less than 64 voxels was excluded.

In the present study, we obtained 49 texture features (17 first order and 32 s order) using the software. The details of the 17 first-order statistics are as follows: seven conventional features (minimum, mean, maximum, standard deviation, first quartile [Q1], second quartile [Q2], third quartile [Q3]); four shape features (volume_ml, volume_voxels, sphericity, and compacity); and six histogram-based features (kurtosis, excesskurtosis, entropy_log10, entropy_log2, energy). Thirty-two second-order statistics consisted of seven gray-level co-occurrence matrix (GLCM)-related features, 11 gray-level run-length matrix (GLRLM)-related features, three neighborhood gray-level different matrix (NGLDM)-related features and 11 gray-level zone length matrix (GLZLM)-related features.

### 2.5. Feature Selection, Modeling, and Statistical Analyses

Univariate analyses including Fisher’s exact test and Mann–Whitney U test were performed. The importance of each feature was independently evaluated according to two different processes. The first process was the evaluation of the degree to which Gini impurity decreased when a feature was chosen to split a node (MDG: Mean Decrease Gini). A feature with higher MDG was supposed to be more important. Parameter tuning using the randomForest and caret packages was applied to RF analysis to optimize the number of trees and the number of features used for each tree with 5-fold cross validation. The model created using MDG evaluation consisted of only a single feature with the highest MDG value. The second process was the Boruta all-relevant feature selection wrapper algorithm using Boruta package [16,17]. The Boruta method enables us to identify relevant features by comparing the importance of each original feature with that of the permuted copies (the three shadow features (shadowMax, shadowMean, shadowMin)) with randomly mixed values. Among the variables selected by the Boruta method, the strength of collinearity between two features was measured by calculating the Spearman’s rank correlation coefficient (Rs). Rs > 0.8 or < −0.8 was determined to indicate very high collinearity. If a pair of features had very high collinearity, the one with the lower importance as judged by the Boruta method was excluded. Among the important features identified by the Boruta method, we finally created the Boruta model by eliminating those with high collinearity. Finally, we performed receiver operating characteristic (ROC) curve analysis to evaluate the performance of classification models based on the two processes, i.e., MDG evaluation and the Boruta method. The optimal cutoff was determined according to Youden’s index. All statistical analyses were performed using R software v4.0.3 (R Foundation for Statistical Computing, Vienna, Austria) and JMP v10.0.2 (SAS Institute Inc., Cary, NC, USA).

## 3. Results

Patient and tumor characteristics are shown in Table 1. The median age of ChRCC patients (*n* = 41) was 60 years, and 17 (41%) were female; the median age of RO patients (*n* = 8) was 60 years, and 4 (50%) were female. Tumor diameters and volumes tended to be larger in ChRCC than RO, although these differences were not significant (median size; 32 vs. 23 mm, *p* = 0.054). The VOI was greater than 64 voxels in all tumors. RO more frequently showed hypointensity in T1-weighted signal intensity (*p* = 0.046), but there were no significant differences between ChRCC and RO in other subjectively assessed MR features.

Parameter tuning for RF analysis showed the minimum error rate of 12.2% with the number of trees (500) and the number of variables (7). RF analysis revealed that the image features of high importance in differentiating ChRCC from RO were ADC-related conventional features. The most important feature in terms of MDG evaluation was the mean ADC value (MDG = 1.282), followed by the Q2_ADC_value (MDG = 1.131) and Q1_ADC_value (MDG = 1.096). The features are listed in order of increasing MDG in Figure 2. The second-order index with the highest MDG was the gray-level run-length matrix short-run high gray-level emphasis (GLRLM_SRHGE, MDG = 0.702).

The 49 extracted features and the three shadow features are listed in Figure 3 in order of importance as determined according to the Boruta method. The Boruta method identified the Q1_ADC_value, mean_ADC_value, Q2_ADC_value, Q3_ADC_value, minimum_ADC_value, gray-level zone length matrix high gray-level zone enhancement (GLZLM_HGZE), gray-level run length matrix high gray-level run enhancement (GLRLM_HGRE), and GLRLM_SRHGE as important features in differentiating ChRCC from RO. Of these, we removed the mean_ADC_value, Q2_ADC_value, Q3_ADC_value, and minimum_ADC_value because the Rs between each of these and the most important feature (Q1_ADC_value) was greater than 0.8 (Figure 4). GLZLM_HGZE remained the second most important feature, and GLRLM_HGRE and GLRLM_SRHGE were removed because the Rs between each of these and GLZLM_HGZE was greater than 0.8. Therefore, the final model consisted of a combination of the Q1_ADC_value and GLZLM_HGZE (labeled as the Boruta model).

The univariate analysis of the mean_ADC_value, Q1_ADC_value, and GLZLM_HGZE, i.e., the components that were identified using the Boruta model as relevant to the differentiation between ChRCC and RO, is shown in Figure 5. The AUC of the mean_ADC_value-only model, which is considered to be the most important for MDG evaluation, was 0.954, and the sensitivity and specificity were 1 and 0.878, respectively, using an ADC value of 1.562 × 10^−3^ mm^2^/s as the cutoff point. The AUC, sensitivity, and specificity of the Boruta model were 0.969, 1, and 0.926, respectively. There was no significant difference between the AUCs of these two models (*p* = 0.236, Figure 6).

## 4. Discussion

Here, we evaluated the usefulness of 3D volumetric data variables obtained from ADC maps using TA in the differentiation of ChRCC from RO. The importance of various image features, including second-order texture features, was assessed using RF-based MDG evaluation and the Boruta method. Both analysis methods indicated that ADC-related conventional features were the most important factors. Radiomics, which focuses on the quantitative information of each voxel and the relationships between voxels that cannot be recognized by the human eye, enables comprehensive image evaluation without subjectivity; to the best of our knowledge, this is the first study to use radiomics to differentiate ChRCC from RO with whole-lesion VOI in ADC maps.

In this study, we focused on identifying the features associated with RF analysis that are important for differentiating between ChRCC and RO. The strength of a decision-tree-based ensemble algorithm such as RF analysis is that it enables us to calculate the importance of each variable. Therefore, we adopted two indices to measure feature importance: the MDG value with RF and the importance with the Boruta method. The MDG value of RF analysis is well-known as a measure of importance, and the strength of the Boruta method, which is also a derivative of RF, lies in its ability to not only rank the importance levels but also to determine significantly important features for the classification. The results of the two evaluation methods were similar in that conventional ADC-related features were at the top of the lists. We found that the mean_ADC_value was the most important based on the MDG value of RF analysis. Furthermore, using the Boruta method and the Spearman’s rank correlation coefficient, we found that the Q1_ADC_value and GLZLM_HGZE were important features that did not have strong collinearity. In the ROC analysis, the Boruta model (the combination of Q1_ADC_value and GLZLM_HGZE) slightly outperformed the mean_ADC_value-only model, but the difference in their performance was not significant. Considering that the Boruta model was optimized for the entire cohort without splitting the data, we could not rule out the possibility that the model was overfitting the cohort. Nonetheless, it is noteworthy that the mean_ADC_value-only model was comparable to the Boruta model in terms of its diagnostic utility in differentiating between ChRCC and RO.

Since the cell density of malignant tumors is higher than that of benign tumors and normal tissues, the ADC value of malignant tumors is lower than that of benign tumors. Furthermore, it has been reported that low ADC values correlate with a high tumor histological grade and high expression of the cell proliferation marker Ki-67 labeling index [8,9]. While the utility of ADC values as an imaging biomarker has previously been reported in studies designed to focus solely on the assessment of ADC values, we would like to emphasize that, in the present study, a comprehensive radiomics analysis identified ADC values as a useful marker for differentiating ChRCC from RO, and that, furthermore, this single representative feature had a discriminatory ability nearly equivalent to that of the Boruta model optimized for the entire cohort.

Quantitative enhancement patterns [18,19,20] and segmental enhancement inversions on CT [21,22] have been investigated to differentiate between ChRCC and RO. However, their usefulness has been inconsistent, and a more accurate and objective measure is needed [22,23]. The usefulness of ADC values in the differential diagnosis of renal tumors has already been evaluated in several studies [11,24,25,26,27,28], and a meta-analysis by Lassel et al. has shown that the ADC values of benign lesions, especially those of RO lesions, are higher than those of malignant lesions [28]. Since ADC values are affected by variation in imaging scanner type and technique, however, it is necessary to compare ChRCC and RO under the same conditions in order to evaluate the usefulness of ADC values. Zhong et al. examined the usefulness of ADC values under the same conditions and reported that the AUC of the ADC value alone calculated from single-slice ROI in distinguishing between ChRCC and RO was as high as 0.931, and that the addition of ADC values to the enhancement ratio did not significantly improve the AUC [12].

As an evaluation based on single-slice ROIs reflects the features of part of a tumor only, whole-lesion VOI analysis that reflects the features of the entire tumor is desirable. In addition, whole-lesion VOI analysis offers the advantage of simplicity in setting the evaluation target, as it does not require the user to avoid necrotic, hemorrhaging, or cystic areas within the tumor. In addition, among the many TFs, the mean ADC value is a well-known imaging feature that can be obtained using common image analysis software. The present study, in which ADC-map-based TA using whole-lesion VOI analysis identified the mean ADC value as an important factor in differentiating ChRCC from RO, is expected to have a significant impact on daily practice.

Our study has several potential limitations. First, this study was a single-center retrospective study with a small sample size. We could not collect enough cases to split the data into a training set and a test set. We positioned this study as a pilot study for feature selection for differentiating ChRCC from RO. The two models we presented in this study were too premature to generalize because they were created by the entire cohort without data splitting, but they were both very simple models consisting of one or two variables, which may reduce the risk of overfitting. A validation study should be performed in an external cohort. Second, bias may have been introduced due to the marked difference in the numbers of patients with ChRCC and RO. Although RO is generally reported to account for about 5% of renal tumors [3], Fujii et al. reported that RO accounted for only 2.8% of patients who underwent partial nephrectomy for presumed RCC in a Japanese dual-center cohort, suggesting that RO might account for a smaller percentage of renal tumors in the Japanese population than it does in Western countries [29]. Third, ChRCC tended to be larger than RO, although there was no significant difference in tumor volume, which may have influenced the results of this study. However, feature extraction was performed in this study based on the same voxel size between the two entities, and tumors smaller than 64 voxels were not included for the analysis. In addition, SHAPE_Volume was not highly significant in differentiating ChRCC from RO in Boruta analysis. Therefore, based on the present analysis using the radiomics analysis of an ADC map, we conclude that SHAPE_Volume was a less important TF in the differentiation between ChRCC and RO. Another inherent limitation was that the outline of each tumor was traced to set the VOI by a single examiner. In order to minimize the possibility of subjectivity, future validations of this method should assess inter-observer agreement.

## 5. Conclusions

Comprehensive analysis of the imaging features obtained by ADC-map-based TA identified the ADC-related conventional features as useful in distinguishing ChRCC from RO. The predictive ability of the mean ADC value was comparable to that of a combination of TFs optimized for the entire cohort under evaluation. The mean ADC value may serve as an important clue in differentiating ChRCC from RO.

## Figures and Tables

**Figure 1 diagnostics-12-00817-f001:**
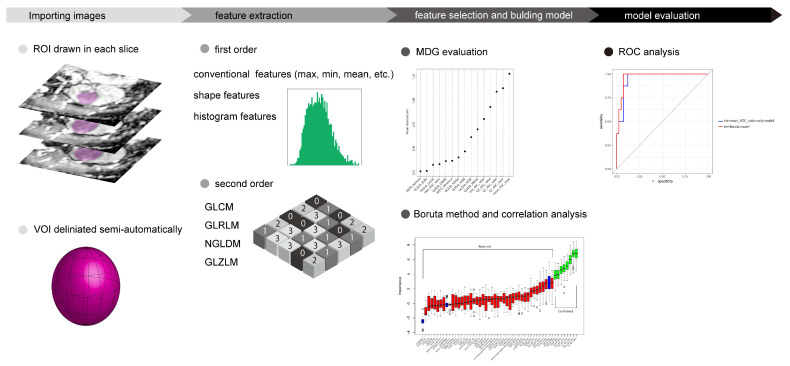
Overview of workflow in this study. Abbreviations: ROI, region of interest; VOI, volume of interest; GLCM, gray-level co-occurrence matrix; GLRLM, gray-level run-length matrix; NGLDM, neighborhood gray-level different matrix; GLZLM, gray-level zone length matrix; MDG, Mean Decrease Gini; ROC, receiver operating characteristic.

**Figure 2 diagnostics-12-00817-f002:**
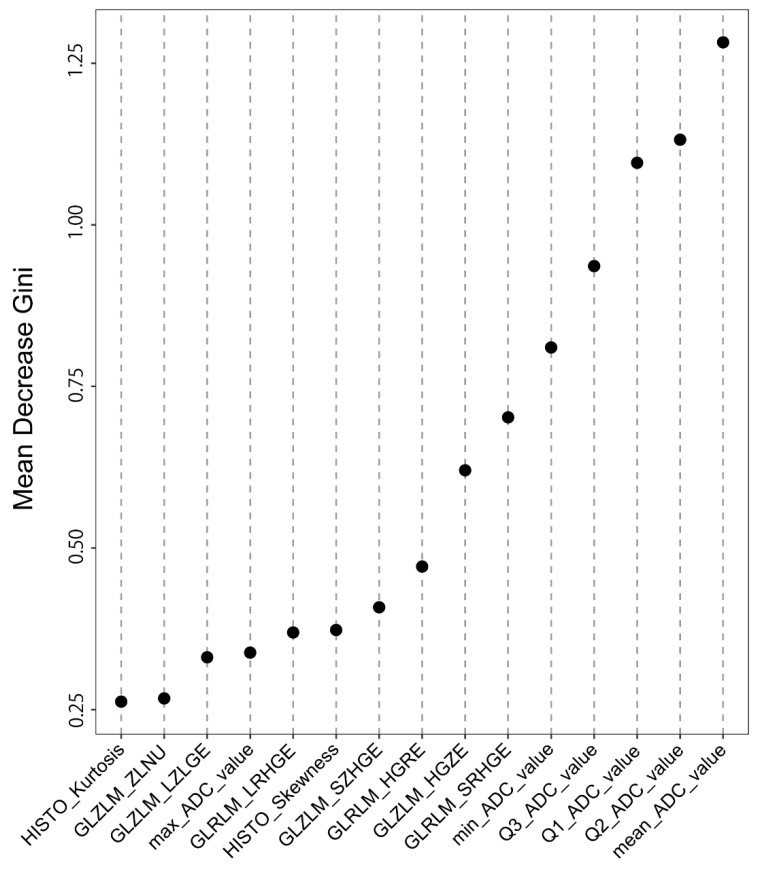
The list of the top 15 features in order of increasing MDG as determined through random forest analysis. The feature with the highest MDG (1.282) was mean_ADC_value. The top five features are all ADC-related conventional features. Abbreviations: MDG, Mean Decrease Gini; ADC, apparent diffusion coefficient.

**Figure 3 diagnostics-12-00817-f003:**
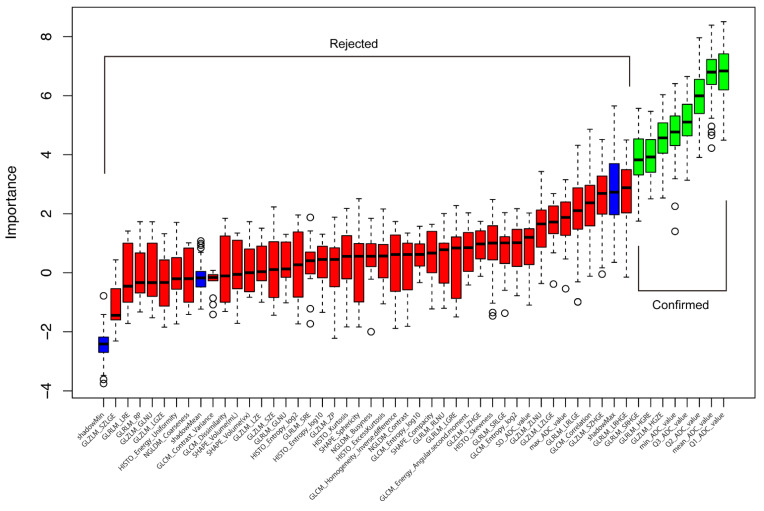
The features listed according to their importance as determined using the Boruta method. The features were color-coded as follows: green for confirmed features; red for rejected features; and blue for shadow features (shadowMax, shadowMean, and shadowMin). The confirmed features were judged to contribute significantly to the differentiation between ChRCC and RO. Abbreviations: ChRCC, chromophobe renal cell carcinoma; RO, renal oncocytoma.

**Figure 4 diagnostics-12-00817-f004:**
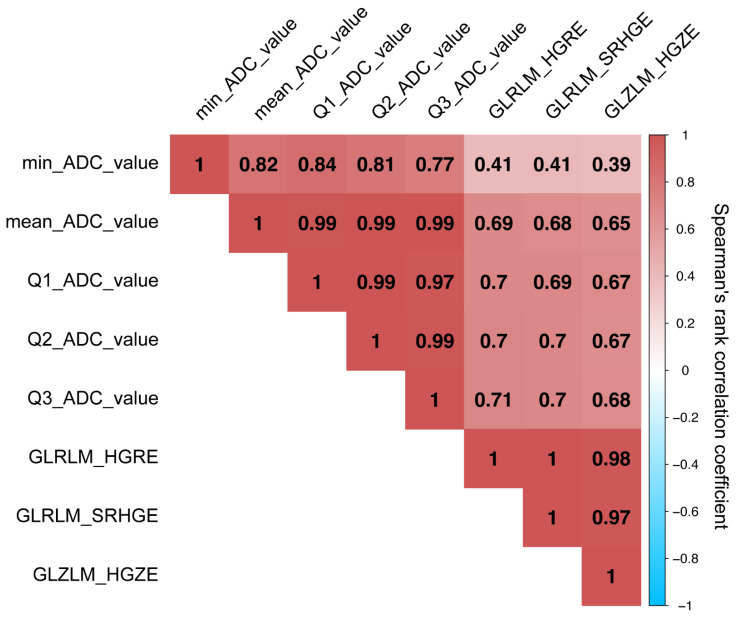
Spearman’s rank correlation coefficient matrix among the eight features judged to be important according to the Boruta method. Each element of the matrix represents the correlation strength between two features: a value closer to 1 indicates a stronger positive correlation (color-coded as red), while a value closer to −1 indicates a stronger negative correlation (color-coded as blue).

**Figure 5 diagnostics-12-00817-f005:**
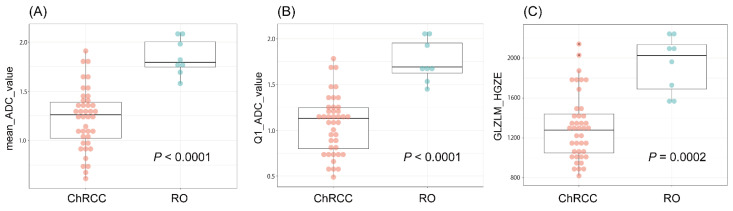
Univariate analysis of three important features that are useful in distinguishing between ChRCC and RO. Box and whiskers represent IQR and range of all values. The units of the vertical axis in (**A**,**B**) are × 10^−3^ mm^2^/s. (**A**) The mean ADC value of ChRCC was significantly lower than that of RO (1.26 × 10^−3^ mm^2^/s vs. 1.79 × 10^−3^ mm^2^/s; *p* < 0.0001). (**B**) The Q1 ADC value of ChRCC was significantly lower than that of RO (1.13 × 10^−3^ mm^2^/s vs. 1.69 × 10^−3^ mm^2^/s; *p* < 0.0001). (**C**) The GLZLM_HGZE of ChRCC was significantly lower that of RO (1.27 × 10^3^ vs. 2.07 × 10^3^; *p* = 0.0002). Abbreviations: ChRCC, chromophobe renal cell carcinoma; RO, renal oncocytoma; IQR, interquartile range; ADC, apparent diffusion coefficient; Q1, first quartile; GLZLM_HGZE, grey-level zone length matrix high gray-level zone emphasis.

**Figure 6 diagnostics-12-00817-f006:**
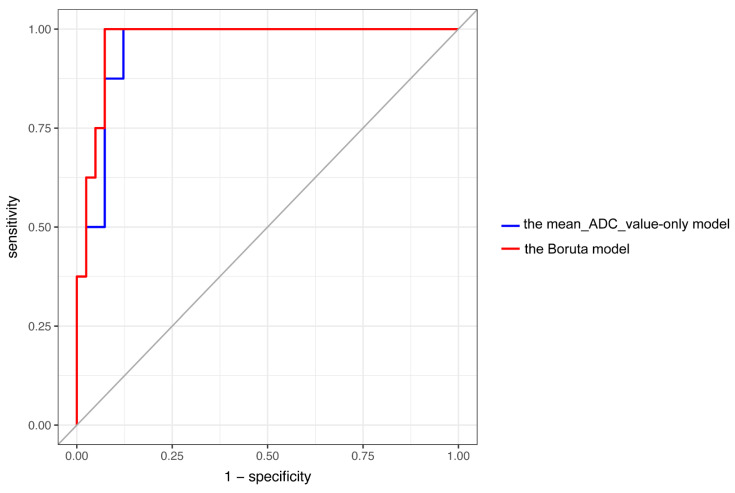
ROC analysis of the mean_ADC_value-only model (blue line) and the Boruta model (red line). The AUC of the mean_ADC_value-only model was 0.954 while that of the Boruta model was 0.969. There was no significant difference between the AUCs of these two models (*p* = 0.236). Abbreviations: ROC, receiver operating characteristic; ADC, apparent diffusion coefficient; AUC: area under the curve.

**Table 1 diagnostics-12-00817-t001:** Characteristics of patients and tumors. The *p* values compared ChRCC with RO and were calculated using Fisher’s exact test (for categorical variables) and the Mann–Whitney U test (for continuous variables). * *p* < 0.05 is considered statistically significant. Abbreviations: ChRCC, chromophobe renal cell carcinoma; RO, renal oncocytoma; IQR, interquartile range; T2WI, T2-weighted image.

Variable		ChRCC (*n* = 41)	RO (*n* = 8)	*p* Value
Age in years, median (IQR)		60 (46–67)	60 (47–72)	0.771
Gender, *n*(%)	Male	24 (58.5)	4 (50.0)	0.710
	Female	17 (41.5)	4 (50.0)	
Laterality, *n*(%)	Left	20 (48.7)	6 (75.0)	0.254
	Right	21 (51.3)	2 (25.0)	
Location, *n*(%)	Upper pole	5 (12.2)	1 (12.5)	1.00
	Middle	29 (70.8)	6 (75.0)	
	Lower pole	7 (17.0)	1 (12.5)	
Tumor diameter (mm), median (IQR)		32 (23–54)	23 (18–25)	0.054
T1-weighted signal intensity, *n*(%)	Hypointense	17 (41.5)	6 (75.0)	0.046 *
	Isointense	17 (41.5)	0(0)	
	Hyperintense	7 (17.0)	2 (25.0)	
T2-weighted signal intensity, *n*(%)	Hypointense	9 (22.0)	0 (0)	0.099
	Isointense	22 (53.6)	3 (37.5)	
	Hyperintense	10 (24.4)	5 (62.5)	
Appearance on T2WI, *n*(%)	Homogeneous	22 (53.6)	3 (37.5)	0.663
	Heterogeneous	19 (46.4)	5 (62.5)	
Margin type, *n*(%)	Well-defined	29 (70.7)	7 (87.5)	0.663
	Indistinct	12 (29.3)	1 (12.5)	
Central scar, *n*(%)	Present	5 (12.2)	1 (12.5)	0.980
	Absent	36 (87.8)	7 (87.5)	
Cystic component, *n*(%)	Present	2 (4.8)	0 (0)	1.00
	Absent	39 (95.2)	8 (100)	
Calcification, *n*(%)	Present	2 (4.8)	0 (0)	1.00
	Absent	39 (95.2)	8 (100)	
Hemorrhage, *n*(%)	Present	3 (7.3)	0(0)	1.00
	Absent	38 (92.7)	8 (100)	
Fat, *n*(%)	Present	1 (2.4)	1 (12.5)	0.302
	Absent	40 (97.6)	7 (87.5)	

## Data Availability

Not applicable.

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
