# Peer review of "Apparent Diffusion Coefficient Map-Based Texture Analysis for the Differentiation of Chromophobe Renal Cell Carcinoma from Renal Oncocytoma"

_diagnostics, 2022, doi:10.3390/diagnostics12040817_

Round 1

Reviewer 1 Report

Dear authors,

the paper is about an interesting topic. It is difficult to distinguish ChRCC from RO on pure imaging features, so every help is welcome. 

I've some concerns about the paper. I'll summarize in the following sentences.

  • The paper need an English revision. Some sentences are difficult to understand. Please also reduce the use of passive voice.
  • Have you also considered the size of the tumor in the model (maybe the shape_volume feature)? I see there is almost significant difference between the mean size of the ChRCC and RO. This, together with the different number of the two groups, may leads to a bias. Smaller tumor is less voxel to analyze, especially in ADC where the voxel size is fairly big. Please add a comment on that.
  • To incresce the reliability of the model, have you considered to add the textural parameters from other sequences, such as T2w and post-contrast T1w images? 
  • About post-contrast T1w images, I see no mention about gadolinium administration. Did you perform or not post contrast acquisition? If no, why?

Please address these concerns.

Best regards

Reviewer 2 Report

This manuscript is evaluating the usefulness of ADC map-based TA in differentiating between ChRCC and RO. This theme is very interesting and very useful for clinical situation, and the general study design is well-developed. But there are some points must be improved, and I think a point (train test splitting) is critical. It must be improved.

I have suggestions.

  1. In line 46, why needle renal biopsy is unreliable? Perhaps the reader should read the reference, but it would be more helpful to them if you clearly state the paper.
  2. Why did you use two different feature selection method? Please specify the reason in the paper. It seems to me that only the results from the Boruta method have been examined in detail.
  3. You should split the dataset training and test portion and the result should be validated by the test portion of the dataset, even if you have small dataset. In addition, it is better to perform K-fold cross validation using only the training portion. The statement in line 277-281 in not a reason not to perform dataset splitting. I think this is critical point. You should be improved.

Reviewer 3 Report

Authors should be congratulated for their interesting work. The topic is still challenging, and it offers a new diagnostic perspective for patients with suspicious renal masses. The manuscript is well-written and easily readable, the methodology is robust, and the tables and figures are clear. Despite the dedication, the paper does not add anything more to the current knowledge (PMID: 33888122).

Round 2

Reviewer 2 Report

Thank you for your revision.

I think your revision is credibly and carefully.

Reviewer 3 Report

Authors should be congratulated for their interesting work. The topic is still challenging, and it offers a new diagnostic perspective for patients with suspicious renal masses. The manuscript is well-written and easily readable, the methodology is robust, and the tables and figures are clear. The authors' explanations are sufficient. The paper can be published.